# Knowledge Distillation based Degradation Estimation for Blind Super-Resolution

**Bin Xia**[1], **Yulun Zhang**[2], **Yitong Wang**[3], **Yapeng Tian**[4],
**Wenming Yang**[1]*, **Radu Timofte**[5], **and Luc Van Gool**[2]
[1]Tsinghua University  [2]ETH Zürich  [3]ByteDance Inc
[4]University of Texas at Dallas  [5]University of Würzburg

## Abstract

Blind image super-resolution (Blind-SR) aims to recover a high-resolution (HR) image from its corresponding low-resolution (LR) input image with unknown degradations. Most of the existing works design an explicit degradation estimator for each degradation to guide SR. However, it is infeasible to provide concrete labels of multiple degradation combinations (*e.g.*, blur, noise, jpeg compression) to supervise the degradation estimator training. In addition, these special designs for certain degradation, such as blur, impedes the models from being generalized to handle different degradations. To this end, it is necessary to design an implicit degradation estimator that can extract discriminative degradation representation for all degradations without relying on the supervision of degradation ground-truth. In this paper, we propose a Knowledge Distillation based Blind-SR network (KDSR). It consists of a knowledge distillation based implicit degradation estimator network (KD-IDE) and an efficient SR network. To learn the KDSR model, we first train a teacher network: KD-IDE$_T$. It takes paired HR and LR patches as inputs and is optimized with the SR network jointly. Then, we further train a student network KD-IDE$_S$, which only takes LR images as input and learns to extract the same implicit degradation representation (IDR) as KD-IDE$_T$. In addition, to fully use extracted IDR, we design a simple, strong, and efficient IDR based dynamic convolution residual block (IDR-DCRB) to build an SR network. We conduct extensive experiments under classic and real-world degradation settings. The results show that KDSR achieves SOTA performance and can generalize to various degradation processes. The code is available at Github.

## 1 Introduction

Single image super-resolution (SISR) aims to recover details of a high-resolution (HR) image from its low-resolution (LR) counterpart, which has a variety of downstream applications (Dong et al., 2014; Zhang et al., 2019; Xia et al., 2022d; Fritsche et al., 2019; Xia et al., 2022c;b). These state-of-the-art methods (Kim et al., 2016; Lim et al., 2017; Lai et al., 2017; Xia et al., 2022a; Wang et al., 2018b) usually assume that there is an ideal bicubic downsampling kernel to generate LR images. However, this simple degradation is different from more complex degradations existing in real-world LR images. This degradation mismatch will lead to severe performance drops.

To address the issue, blind super-resolution (Blind-SR) methods are developed. Some Blind-SR works (Wang et al., 2021a; Luo et al., 2022) use the classical image degradation process, given by Eq. 1. Recently, some works (Cai et al., 2019; Bulat et al., 2018) attempted to develop a new and complex degradation process to better cover real-world degradation space, which forms a variant of Blind-SR called real-world super-resolution (Real-SR). The representative works include BSR-GAN (Zhang et al., 2021) and Real-ESRGAN (Wang et al., 2021b), which introduce comprehensive degradation operations such as blur, noise, down-sampling, and JPEG compression, and control the severity of each operation by randomly sampling the respective hyper-parameters. To better simulate the complex degradations in real-world, they also apply random shuffle of degradation orders (Zhang et al., 2021) and second-order degradation (Wang et al., 2021b) respectively.

Since Blind-SR faces almost infinite degradations, introducing prior degradation information to SR networks can help to constrain the solution space and boost SR performance. As shown in Fig. 1, the way to obtain degradation information can be divided into three categories: **(1)** Several Non-Blind SR methods (Zhang et al., 2018a; Shocher et al., 2018; Zhang et al., 2020; Soh et al., 2020; Xu et al., 2020) directly take the known degradation information as prior (Fig. 1 (a)). **(2)** Most of

---

*Corresponding Author: Wenming Yang, yang.wenming@sz.tsinghua.edu.cn

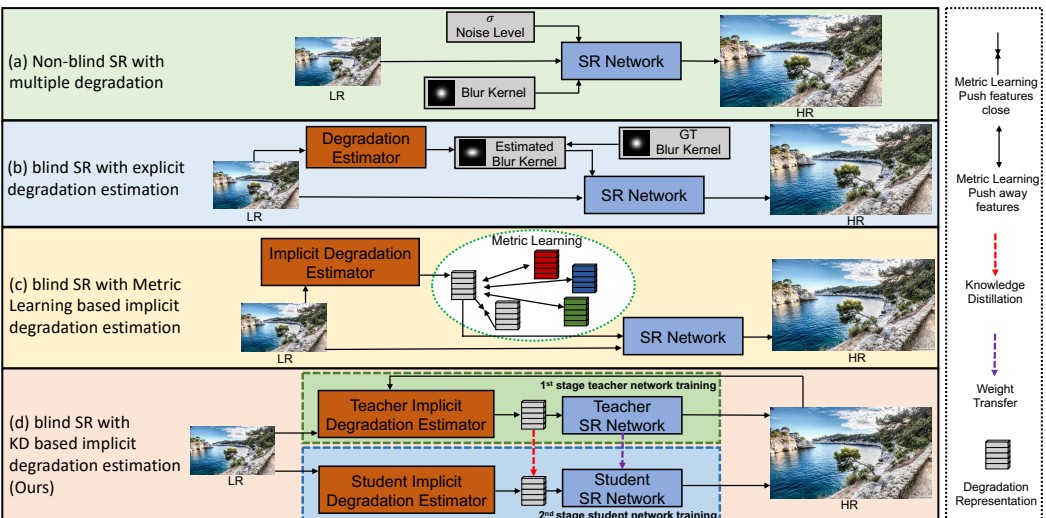

Figure 1: The illustration of different degradation estimators. (a) Non-blind SR methods directly use known degradation information to guide SR networks, such as SRMD (Zhang et al., 2018a). (b) Many Blind-SR methods estimate the explicit degradation with the supervision of ground-truth degradation. (c) Several methods use metric learning to distinguish degradation roughly. (d) Our knowledge distillation (KD) based implicit degradation estimator can estimate accurate implicit degradation representation to guide SR without ground-truth degradation supervision.

Blind-SR methods (Gu et al., 2019; Luo et al., 2020; Wang et al., 2021a; Liang et al., 2022; Luo et al., 2022) adopt explicit degradation estimators, which are trained with ground-truth degradation (Fig. 1 (b)). However, these explicit degradation estimators are elaborately designed for specific degradation processes. The specialization makes them hard to transfer to handle other degradation processes. In addition, it is challenging to annotate precise ground-truth labels to represent the multiple degradation combination (Zhang et al., 2021; Wang et al., 2021b) for supervised degradation learning. Therefore, developing implicit degradation representation (IDR) based methods is important. **(3)** Recently, as shown in Fig. 1 (c), DASR (Wang et al., 2021a) and MM-RealSR (Mou et al., 2022) use metric learning to estimate IDR and quantize degradation severity respectively. However, metric learning methods roughly distinguish degradations by pushing away or pulling close features, which is unstable and cannot fully capture discriminative degradation characteristics for Blind-SR.

In this paper, we aim to design an efficient implicit degradation representation (IDR) learning SR framework that can easily adapt to any degradation process. To this end, we develop a novel knowledge distillation based Blind-SR Network (KDSR). Specifically, as shown in Fig. 1 (d), KDSR uses a knowledge distillation based implicit degradation estimator (KD-IDE) to predict accurate IDR. Furthermore, we propose a strong yet efficient SR network based on our newly developed IDR based Dynamic Convolution Residual Blocks (IDR-DCRB) to reconstruct the HR image with the guidance of IDR. For the training process, we first input HR and LR images to the teacher KD-IDE$_T$, which is optimized with the SR network together. Given the paired HR and LR images, teacher KD-IDE$_T$ can easily extract the latent degradation information in LR images. Then, we use a student KD-IDE$_S$ to learn to extract the same IDR as that of KD-IDE$_T$ from LR images directly. Extensive experiments can demonstrate the effectiveness of the proposed KDSR. Our main contributions are threefold:

- We propose KDSR, a strong, simple, and efficient baseline for Blind-SR, can generalize to any degradation process, which addresses the weakness of explicit degradation estimation.

- We propose a novel KD based implicit degradation representation (IDR) estimator. To the best of our knowledge, the design of IDR estimation has received little attention so far. Besides, we propose an efficient IDR-based SR network to fully utilize IDR to guide SR.

- Extensive experiments show that the proposed KDSR can achieve excellent Blind-SR performance in different degradation settings from simple to complex.

## 2 RELATED WORK

### 2.1 BLIND SUPER-RESOLUTION

In the past few years, numerous Non-Blind SISR methods (Dong et al., 2014; Lim et al., 2017; Zhang et al., 2018a; Ledig et al., 2017; Johnson et al., 2016; Ma et al., 2020; Xia et al., 2023) have

achieved promising performance and have been widely studied. However, the performance of these methods will dramatically decline as there is a degradation gap between training and test data. As a remedy, SRMD (Zhang et al., 2018a), USRNet (Zhang et al., 2020) and some other methods (Zhang et al., 2019; Xu et al., 2020; Shocher et al., 2018; Soh et al., 2020) utilize the blur kernel and noise level as additional input. Although these methods can deal with multiple degradations with a single model, they require accurate degradation estimation, which is also a challenging task.

To handle unknown degradation, a few Blind-SR methods have been proposed. Some methods, such as IKC (Gu et al., 2019) and DAN (Luo et al., 2020), use the classical Blind-SR degradation process and combine a blur kernel estimator with SR networks, which can be adaptive to images degraded from various blur kernels (Kim et al., 2021; Cornillere et al., 2019). Besides, KMSR (Zhou & Susstrunk, 2019) constructs a kernel pool by utilizing a generative adversarial network (Goodfellow et al., 2014). Recently, some works like BSRGAN (Zhang et al., 2021) and Real-ESRGAN (Wang et al., 2021b) design more complex and comprehensive degradation processes to better cover the real-world degradation space, which becomes a variant of Blind-SR called Real-SR.

For each degradation type and process, previous Blind-SR methods (Gu et al., 2019; Liang et al., 2022) tend to specially design an explicit degradation estimator. That is quite complex and hard to provide ground-truth labels for multiple degradation combinations. Recently, DASR (Wang et al., 2021a) and MM-RealSR (Mou et al., 2022) use metric learning to roughly distinguish various degradations, which is not accurate enough to provide degradation representation to guide SR. In this paper, we propose to estimate IDR accurately and fully use it for SR in an efficient way.

## 2.2 KNOWLEDGE DISTILLATION

The purpose of knowledge distillation (KD) (Hinton et al., 2015) is to transfer the representation ability of a teacher network to a student network. KD has been widely used to compress models, typically for classification tasks. Specifically, they (Ahn et al., 2019) compress the classification models by enforcing the output distribution between the teacher and student networks to be close. Recently, some works extend KD to feature distillation, such as intermediate feature learning (Romero et al., 2014) and pairwise relations in intermediate feature learning (Liu et al., 2019). For the SISR task, SRKD (Gao et al., 2018) and FAKD (He et al., 2020) apply the KD between intermediate feature maps to compress models. To obtain more efficient SR networks, PISR (Lee et al., 2020) further introduces variational information distillation (Ahn et al., 2019) to maximize the mutual information between intermediate feature maps of teacher and student SR networks. Different from previous works adopting KD for model compression, we develop KDSR to obtain accurate IDR.

## 3 METHODOLOGY

### 3.1 OVERVIEW

Blind SR methods can be summarized two categories: classic blind-SR and real-world blind-SR.

*For classic blind-SR*, some Blind-SR works (Wang et al., 2021a; Luo et al., 2022) use the classical image degradation process, given by

$$\mathbf{y} = (\mathbf{x} \otimes \mathbf{k}) \downarrow_s + \mathbf{n}, \tag{1}$$

where $\otimes$ denotes convolution operation. $\mathbf{x}$ and $\mathbf{y}$ are HR and corresponding LR images respectively. $\mathbf{k}$ is blur kernel and $\mathbf{n}$ is additional white Gaussian noise. $\downarrow_s$ refers to downsampling operation with scale factor $s$. The severity of blur and noise are unknown, which are randomly sampling the respective hyper-parameters to adjust severity and form almost infinite degradation space.

Given an input LR image $\mathbf{y}$ and applied blur kernel $\mathbf{k}$, the classic blind-SR methods (Gu et al., 2019; Luo et al., 2020; 2022) pretrain an explicit degradation estimator to estimate the blur kernel applied on $\mathbf{y}$ with the supervision of groundtruth $\mathbf{k}$. Then, their SR network can use the estimated blur kernel to perform SR on LR image $\mathbf{y}$. The SR network is trained with loss function $\mathcal{L}_{rec}$.

$$\mathcal{L}_{rec} = \|I_{HR} - I_{SR}\|_1, \tag{2}$$

where $I_{HR}$ and $I_{SR}$ are real and SR images separately.

*Real-world blind-SR* is a variant of classic blind-SR, in which more complicated degradation is adopted. The real-world blind-SR approaches (Wang et al., 2021b; Zhang et al., 2021) introduce comprehensive degradation operations such as blur, noise, down-sampling, and JPEG compression, and control the severity of each operation by randomly sampling the respective hyper-parameters. Moreover, they apply random shuffle of degradation orders and second-order degradation to increase

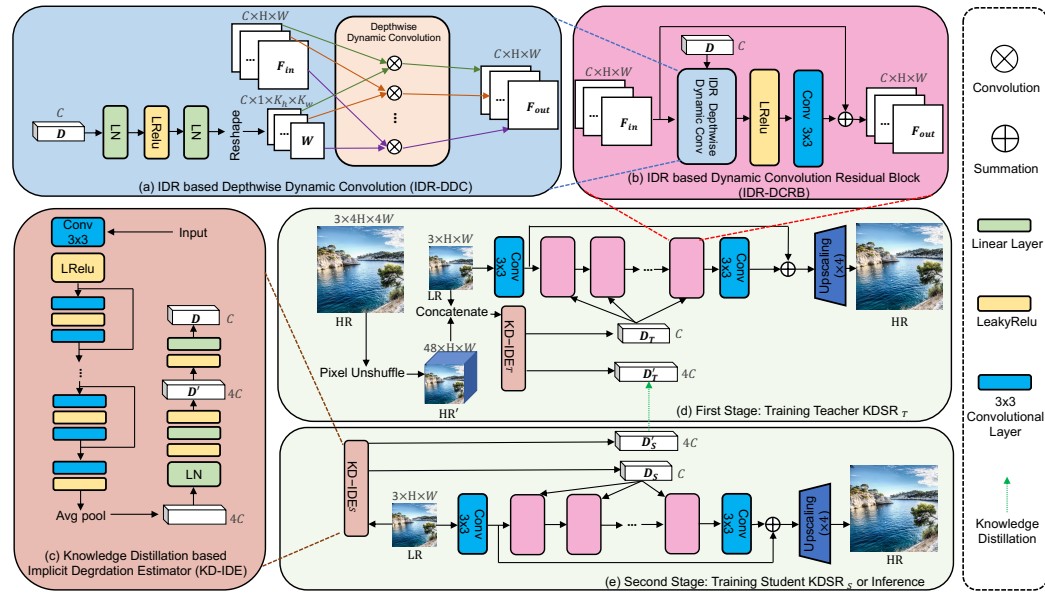

Figure 2: The overview of our proposed knowledge distillation based Blind-SR network (KDSR), which consists of a KD based implicit degradation estimator network (KD-IDE) and a SR network mainly formed by the IDR based Depthwise Dynamic convolution (IDR-DDC).

degradation complexity. Since degradation is complex and cannot provide specific degradation labels, they directly use SR networks without degradation estimators. Their SR networks emphasize visual quality trained with $\mathcal{L}_{vis}$.

$$\mathcal{L}_{vis} = \lambda_{rec}\mathcal{L}_{rec} + \lambda_{per}\mathcal{L}_{per} + \lambda_{adv}\mathcal{L}_{adv}, \tag{3}$$

where $\mathcal{L}_{per}$ and $\mathcal{L}_{adv}$ are perceptual (Johnson et al., 2016) and adversarial loss (Wang et al., 2021b).

As shown in Fig. 2, we propose a KDSR, consisting of KD-IDE and an efficient SR network. Different previous explicit degradation estimation based blind-SR methods Gu et al. (2019); Luo et al. (2020; 2022); Liang et al. (2022), our KD-IDE does not requires degradation labels for training, which can generalize to any degradation process. Moreover, our the design of our SR network is neat and efficient, which is practical and can fully use degradation information for SR. There are two stage training processes for KDSR, including Teacher KDSR$_T$ and Student KDSR$_S$ training. We first train the KDSR$_T$: we input the paired HR and LR images to the KD-IDE$_T$ and obtain the implicit degradation representation (IDR) $\mathbf{D}_T$ and $\mathbf{D}'_T$, where $\mathbf{D}_T$ is used to guide the SR. After that, we move on to KDSR$_S$ training. We initialize the KDSR$_S$ with the KDSR$_T$'s parameters and make the KDSR$_S$ learn to directly extract $\mathbf{D}'_S$ same as $\mathbf{D}'_T$ from LR images.

## 3.2 Knowledge Distillation based Implicit Degradation Estimator

Most Blind-SR methods elaborately design an explicit degradation estimator for each degradation type and process. There are several limitations for explicit degradation estimators: **(1)** These special designs for specific degradation processes make the explicit estimator hard to be transferred to other degradation settings. **(2)** It is complex to provide various degradation labels for explicit degradation estimator training, especially the random combination of multiple degradations (Wang et al., 2021b). Therefore, we develop a KD based implicit degradation estimator (KD-IDE), which can distinguish various degradations accurately without the supervision of degradation ground-truth.

As shown in Fig. 2 (c), we can divide KD-IDE into several parts: **(1)** We take the LR images and the concatenation of LR and HR images as input for KD-IDE$_S$ and KD-IDE$_T$, respectively. Specially, for the KD-IDE$_T$ (Fig. 2 (d)), it can easily extract the degradation which makes HR degrade to LR images by providing paired HR and LR images and jointly being optimized with the SR network. Since there is a spatial size difference between HR and LR images, we perform the Pixel-Unshuffle operation on HR images $I_{HR} \in \mathbb{R}^{3 \times 4H \times 4W}$ to be $I_{HR'} \in \mathbb{R}^{48 \times H \times W}$ and then concatenate it with LR images to obtain an input $I \in \mathbb{R}^{51 \times H \times W}$. **(2)** The input passes through the first convolution to become feature maps. It is noticeable that the input channels in the first convolution are 3 and 51 for KD-IDE$_S$ and KD-IDE$_T$, respectively. **(3)** After that, we use numerous residual blocks to further extract features and obtain a rough degradation vector by Average Pooling operation. **(4)** We use the two linear layers to refine the degradation vector and obtain IDR $\mathbf{D}' \in \mathbb{R}^{4C}$, which is used for KD. **(5)** Although $\mathbf{D}'$ has $4C$ channels to accurately present degradation and can give

more degradation information for the student network to learn, it will consume a large number of computational resources used in IDR-DDC. Hence, we need to further compress it with a linear layer and obtain another IDR $\mathbf{D} \in \mathbb{R}^C$ to guide SR. More details of KD training are given in Sec. 3.4.

### 3.3 IMAGE SUPER-RESOLUTION NETWORK

As for the design of SR network, we should consider three points: **(1)** After we obtain the IDR, it is important to design a SR network that can fully use the estimated degradation prior for SR. **(2)** An ideal Blind-SR network is likely to be used in practice, the structure of which should be simple. Thus, we also try to make the network formed by one type of simple and strong enough module. **(3)** The huge computation consumption usually limits the application of models, especially on edge devices. Thus, it is necessary to design an efficient model.

As shown in Fig. 2 (a), (b), and (d), our SR network can be divided into three hierarchies. **(1)** We first propose a convolution unit called IDR based Depthwise Dynamic Convolution (IDR-DDC). Motivated by UDVD (Xu et al., 2020), we adopt the dynamic convolution to use IDR to guide SR. Specifically, to fully use the estimated IDR, as displayed in Fig. 2 (a), we generate specific convolution weights according to the IDR $\mathbf{D}$. However, if we generate ordinary convolution weights, the computational cost will be quite large and affect the efficiency of the network. Thus, we further introduce depthwise convolution (Howard et al., 2017), which merely consumes about $\frac{1}{C}$ computation and parameters of ordinary convolution. The IDR-DDC can be mathematically expressed as:

$$\mathbf{W} = \text{Reshape}\left(\phi\left(\mathbf{D}\right)\right), \tag{4}$$

$$\mathbf{F}_{out}[i, :, :] = \mathbf{F}_{in}[i, :, :] \otimes \mathbf{W}[i, :, :, :], i \in [0, C), \tag{5}$$

where $\phi(.)$ and $\otimes$ are two linear layers and convolution operation separately; $\mathbf{D} \in \mathbb{R}^C$ indicates IDR, $\phi\left(\mathbf{D}\right) \in \mathbb{R}^{CK_hK_w}$ is the output of $\phi(.)$, $\mathbf{W} \in \mathbb{R}^{C \times 1 \times K_h \times K_w}$ is weights of dynamic convolution ; $\mathbf{F}_{in}$ and $\mathbf{F}_{out} \in \mathbb{R}^{C \times H \times W}$ are input and output feature maps respectively. **(2)** As shown in Fig. 2 (b), motivated by EDSR (Lim et al., 2017), we develop IDR based Dynamic Convolution Residual Blocks (IDR-DCRB) to realize deep model. For the first convolution of IDR-DCRB, we use the IDR-DDC to utilize degradation information. However, IDR-DDC lacks interaction between different channels. Thus, we adopt ordinary convolution as the second convolution. **(3)** For simplicity, as shown in Fig. 2 (d) or (e), we mainly stack the IDR-DCRB to form the SR network.

### 3.4 TRAINING PROCESS

KDSR has a two-stage training process. **(1)** As shown in the Fig. 2 (d), we first train the teacher KDSR$_T$. we input the paired LR and HR images to the KD-IDE$_T$ obtain the IDR $\mathbf{D_T}$ and $\mathbf{D'_T}$. Then, $\mathbf{D_T}$ is used to generate specific degradation weights for dynamic convolution. After that, the specific degradation SR network will restore the LR images. By jointly optimizing the teacher SR network and KD-IDE$_T$ with the $\mathcal{L}_1$ Loss (Eq. 2), the KD-IDE can effectively extract accurate IDR to guide SR network. **(2)** After finishing KDSR$_T$ training, we move on to train KDSR$_S$. As shown in the Fig. 2 (e), different from KD-IDE$_T$, we only input the LR images to the KD-IDE$_S$, obtaining IDR $\mathbf{D_S}$ and $\mathbf{D'_S}$. The other steps are the same as KDSR$_T$ training except for the adopted loss functions. Specifically, we introduce a knowledge distillation (KD) function (Eq. 6) to enforce the KD-IDE$_S$ directly extracting the same accurate IDR as KD-IDE$_T$ from LR images. In addition, for the classic degradation model (Eq. 1), following previous Blind-SR works (Gu et al., 2019; Wang et al., 2021a), we adopt $\mathcal{L}_{rec}$ (Eq. 2) and can set the total loss function as $\mathcal{L}_{classic}$ (Eq. 7). For more complex degradation processes (Real-SR), following $\mathcal{L}_{vis}$ (Eq. 3) of Real-ESRGAN (Wang et al., 2021b), we propose $\mathcal{L}_{real}$ (Eq. 8). More details are given in appendix.

$$\mathcal{L}_{kl} = \sum_{j=[0,4C)} \mathbf{D}'_{Tnorm}(j) \log\left(\frac{\mathbf{D}'_{Tnorm}(j)}{\mathbf{D}'_{Snorm}(j)}\right), \tag{6}$$

$$\mathcal{L}_{classic} = \lambda_{rec}\mathcal{L}_{rec} + \lambda_{kl}\mathcal{L}_{kl}, \tag{7}$$

$$\mathcal{L}_{real} = \lambda_{rec}\mathcal{L}_{rec} + \lambda_{kl}\mathcal{L}_{kl} + \lambda_{per}\mathcal{L}_{per} + \lambda_{adv}\mathcal{L}_{adv}, \tag{8}$$

where $\mathbf{D}'_{Tnorm}$ and $\mathbf{D}'_{Snorm}$ are normalized with softmax operation of $\mathbf{D}'_T$ and $\mathbf{D}'_S$ separately. $\mathcal{L}_{per}$ and $\mathcal{L}_{adv}$ are perceptual and adversarial loss. $\lambda_{kl}$, $\lambda_{per}$ and $\lambda_{adv}$ denote the balancing parameters.

## 4 EXPERIMENTS

### 4.1 SETTINGS

We train and test our method on classic and real-world degradation settings. For the *classic degradation*, following previous works (Gu et al., 2019; Luo et al., 2022), we combine

Table 1: 4× SR quantitative comparison on datasets with Gaussian8 kernels. The bottom three methods marked in rouse use IDR to guide blind SR. The FLOPs and runtime are computed based on an LR size of $180 \times 320$. Best and second best performance are in red and blue colors, respectively.

| Methods | Param (M) | FLOPs (G) | Time (ms) | Set5 PSNR | Set5 SSIM | Set14 PSNR | Set14 SSIM | BSD100 PSNR | BSD100 SSIM | Urban100 PSNR | Urban100 SSIM | Manga109 PSNR | Manga109 SSIM |
|---|---|---|---|---|---|---|---|---|---|---|---|---|---|
| Bicubic | - | - | - | 24.57 | 0.7108 | 22.79 | 0.6032 | 23.29 | 0.5786 | 20.35 | 0.5532 | 21.50 | 0.6933 |
| RCAN | 15.59 | 1082.41 | 556.21 | 26.60 | 0.7598 | 24.85 | 0.6513 | 25.01 | 0.6170 | 22.19 | 0.6078 | 23.52 | 0.7428 |
| Bicubic+ZSSR | 0.23 | - | 30946.60 | 26.45 | 0.7279 | 24.78 | 0.6268 | 24.97 | 0.5989 | 21.11 | 0.5805 | 23.53 | 0.724 |
| IKC | 5.32 | 2528.03 | 1053.79 | 31.67 | 0.8829 | 28.31 | 0.7643 | 27.37 | 0.7192 | 25.33 | 0.7504 | 28.91 | 0.8782 |
| DANv1 | 4.33 | 1098.33 | 201.04 | 31.89 | 0.8864 | 28.42 | 0.7687 | 27.51 | 0.7248 | 25.86 | 0.7721 | 30.50 | 0.9037 |
| DANv2 | 4.71 | 1088.14 | 201.51 | 32.00 | 0.8885 | 28.50 | 0.7715 | 27.56 | 0.7277 | 25.94 | 0.7748 | 30.45 | 0.9037 |
| AdaTarget | 16.70 | 1032.59 | 109.77 | 31.58 | 0.8814 | 28.14 | 0.7626 | 27.43 | 0.7216 | 25.72 | 0.7683 | 29.97 | 0.8955 |
| DCLS | 13.63 | - | 175.84 | 32.12 | 0.8890 | 28.54 | 0.7728 | 27.60 | 0.7285 | 26.15 | 0.7809 | 30.86 | 0.9086 |
| DASR | 5.84 | 185.66 | 44.14 | 31.46 | 0.8789 | 28.11 | 0.7603 | 27.44 | 0.7214 | 25.36 | 0.7506 | 29.39 | 0.8861 |
| KDSR$_S$-M (Ours) | 5.80 | 191.42 | 38.74 | 32.02 | 0.8892 | 28.46 | 0.7761 | 27.52 | 0.7281 | 25.96 | 0.7760 | 30.58 | 0.9026 |
| KDSR$_S$-L (Ours) | 14.19 | 623.61 | 149.14 | 32.11 | 0.8933 | 28.68 | 0.7867 | 27.64 | 0.7300 | 26.15 | 0.7830 | 30.99 | 0.9069 |

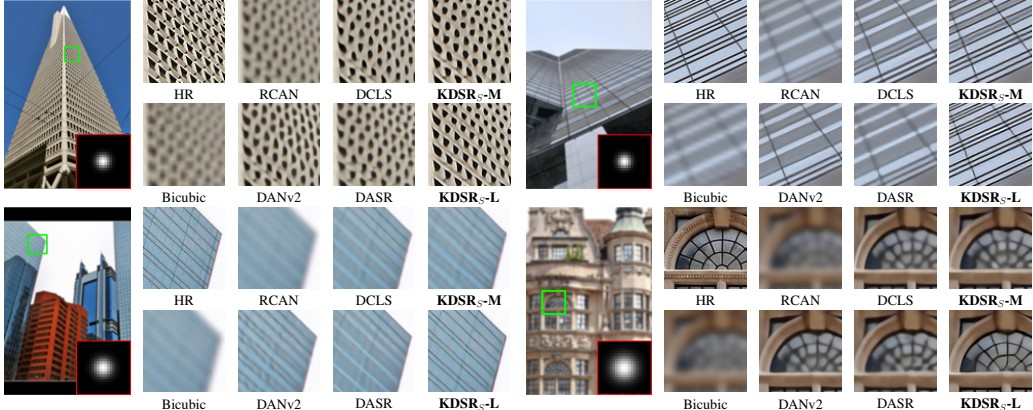

Figure 3: Visual comparison (4×) of Blind-SR methods on isotropic Gaussian kernels.

800 images in DIV2K (Agustsson & Timofte, 2017) and 2,650 images in Flickr2K (Timofte et al., 2017) as the DF2K training set. The batch sizes are set to 64, and the LR patch sizes are 64×64. We use Adam optimizer with $\beta_1 = 0.9$, $\beta_2 = 0.99$. We train both teacher and student networks with 600 epochs and set their initial learning rate to $10^{-4}$ and decrease to half after every 150 epochs. The loss coefficient $\lambda_{rec}$ and $\lambda_{kd}$ are set to 1 and 0.15 separately. The SR results are evaluated with PSNR and SSIM on the Y channel in the YCbCr space. **(1)** In Sec. 4.2, we train and test on isotropic Gaussian kernels following the setting in Gu et al. (2019). Specifically, the kernel sizes are fixed to 21×21. In training, the kernel width $\sigma$ ranges are set to [0.2, 4.0] for scale factors 4. We uniformly sample the kernel width in the above ranges. For testing, we adopt the Gaussian8 (Gu et al., 2019) kernel setting to generate evaluation datasets. Gaussian8 uniformly chooses 8 kernels from range [1.80, 3.20] for scale 4. The LR images are obtained by blurring and downsampling the HR images with selected kernels. **(2)** In Sec. 4.3, we also validate our methods on anisotropic Gaussian kernels and noises following the setting in (Wang et al., 2021a). Specifically, We set the kernel size to 21×21 for scale factor 4. In training, we use the additive Gaussian noise with covariance $\sigma = 25$ and adopt anisotropic Gaussian kernels characterized by Gaussian probability density function $N(0, \Sigma)$ with zero mean and varying covariance matrix $\Sigma$. The covariance matrix $\Sigma$ is determined by two random eigenvalues $\lambda_1, \lambda_2 \sim U(0.2, 4)$ and a random rotation angle $\theta \sim U(0, \pi)$.

For the *real-world degradation*, in Sec. 4.4, similar to Real-ESRGAN (Wang et al., 2021b), we adopt DF2K and OutdoorSceneTraining (Wang et al., 2018a) datasets for training. We set the learning rate of the KDSR$_T$ to $2 \times 10^{-4}$ and pre-train it with only Eq. 2 by 1000K iterations. Then, we optimize KDSR$_S$ with Eq. 7 by 1000K iterations and continue to train it with Eq. 8 by 400K iterations. The learning rate is fixed as $10^{-4}$. For optimization, we use Adam with $\beta_1 = 0.9$, $\beta_2 = 0.99$. In both two stages of training, we set the batch size to 48, with the input patch size being 64.

## 4.2 EVALUATION WITH ISOTROPIC GAUSSIAN KERNELS

We first evaluate our KDSR on degradation with isotropic Gaussian kernels. We compare the KDSR with several SR methods, including RCAN (Zhang et al., 2018c), ZSSR (Shocher et al., 2018),

Table 2: PSNR results achieved on Set14 (Zeyde et al., 2010) under anisotropic Gaussian blur and noises. The bottom two methods marked in rouse use IDR to guide blind SR. The best results are marked in bold. The runtime is measured on an LR size of $180 \times 320$.

| Method | Params | Time | Noise $\sigma$ | Blur Kernel | | | | | | | | |
|---|---|---|---|---|---|---|---|---|---|---|---|---|
| DnCNN + RCAN | 650K+15.59M | 556.21ms | 0 | 24.28 | 24.47 | 24.6 | 24.64 | 24.58 | 24.47 | 24.31 | 23.97 | 23.01 |
| | | | 10 | 23.88 | 24.03 | 24.16 | 24.21 | 24.13 | 24.03 | 23.88 | 23.62 | 22.76 |
| | | | 20 | 23.45 | 23.58 | 23.70 | 23.73 | 23.69 | 23.57 | 23.42 | 23.23 | 22.46 |
| DnCNN +IKC | 650K+5.32M | 1053.79ms | 0 | 24.76 | 25.55 | 26.54 | 27.33 | 26.55 | 25.55 | 24.64 | 25.99 | 25.49 |
| | | | 10 | 24.20 | 24.54 | 24.86 | 24.96 | 24.78 | 24.52 | 24.23 | 24.19 | 23.14 |
| | | | 20 | 23.62 | 23.87 | 24.07 | 24.15 | 24.06 | 23.86 | 23.65 | 23.59 | 22.71 |
| DnCNN +DCLS | 650K+19.05M | 192.83ms | 0 | 25.80 | 26.20 | 26.45 | 26.46 | 26.30 | 26.20 | 26.39 | 25.57 | 23.96 |
| | | | 10 | 24.05 | 24.28 | 24.44 | 24.50 | 24.40 | 24.27 | 24.09 | 23.85 | 22.90 |
| | | | 20 | 23.58 | 23.75 | 23.88 | 23.93 | 23.88 | 23.72 | 23.56 | 23.40 | 22.58 |
| DASR | 5.84M | 44.14ms | 0 | 27.20 | 27.62 | 27.74 | 27.85 | 27.82 | 27.62 | 27.38 | 27.44 | 26.27 |
| | | | 10 | 25.26 | 25.57 | 25.64 | 25.69 | 25.62 | 25.54 | 25.42 | 25.20 | 24.37 |
| | | | 20 | 23.68 | 23.87 | 24.20 | 24.32 | 24.09 | 23.91 | 23.76 | 23.81 | 22.87 |
| KDSR$_S$ (Ours) | 5.80M | 38.74ms | 0 | **27.67** | **27.99** | **28.14** | **28.20** | **28.12** | **27.99** | **27.80** | **27.87** | **26.52** |
| | | | 10 | **25.74** | **25.91** | **25.97** | **26.00** | **25.96** | **25.88** | **25.75** | **25.50** | **24.67** |
| | | | 20 | **24.72** | **24.89** | **24.92** | **24.89** | **24.92** | **24.82** | **24.70** | **24.59** | **23.84** |

Figure 4: $4\times$ visual comparison. Noise levels are set to 10 and 20 for these two images separately. IKC (Gu et al., 2019), DAN (Luo et al., 2020), AdaTarget (Jo et al., 2021), and DASR (Wang et al., 2021a). Note that RCAN is a state-of-the-art SR method for bicubic degradation. For a fair comparison on different model sizes, we develop KDSR$_S$-M and KDSR$_S$-L by adjusting the depth and channels of the network. We apply Gaussian8 (Gu et al., 2019) kernel setting on five datasets, including Set5 (Bevilacqua et al., 2012), Set14 (Zeyde et al., 2010), B100 (Martin et al., 2001), Urban100 (Huang et al., 2015), and Manga109 (Matsui et al., 2017), to generate evaluation datasets.

The quantitative results are shown in Tab. 1. We can see that our KDSR$_S$-M surpasses DASR by 0.6dB, 0.39dB, 0.67dB and 1.24dB on Set5, Set14, Urban100 and Manga109 datasets separately. In addition, compared with the Blind-SR method DANv2, our KDSR$_S$-M achieves better performance consuming only 21% FLOPs of DANv2. It is because that DANv2 uses an iterative strategy to estimate accurate explicit blur kernels, which requires many computations. Besides, compared with the SOTA Blind-SR method DCLS, our KDSR$_S$-L achieves better performance on almost all datasets consuming less time. It is notable that DCLS specially designed an explicit degradation estimator for blur kernel, while the KD-IDE in our KDSR is simple and can adapt to any degradation process. The qualitative results are shown in Fig. 3. We can see that our KDSR$_S$-L has more clear textures compared with other methods. Our KDSR$_S$-M also achieves better visual results than DANv2.

### 4.3 EVALUATION WITH ANISOTROPIC GAUSSIAN KERNELS AND NOISES

We evaluate our KDSR on degradation with anisotropic Gaussian kernels and noises by adopting 9 typical blur kernels and different noise levels. We compare our KDSR with SOTA blind-SR methods, including RCAN (Zhang et al., 2018c), IKC (Gu et al., 2019), DCLS (Luo et al., 2022) and DASR (Wang et al., 2021a). Since RCAN, IKC, and DCLS cannot deal with noise degradation, we use DnCNN (Zhang et al., 2017), a SOTA denoising method, to denoise images for them.

The quantitative results are shown in Tab. 2. Compared with the SOTA explicit degradation estimation based on Blind-SR methods DCLS, our KDSR$_S$ surpasses it by over 1 dB under almost all degradation settings consuming 29.4% parameters and 5.1% runtime. Furthermore, as $\sigma = 20$, our KDSR$_S$ surpasses DASR about 1dB with less parameters and runtime. This shows the superiority of knowledge distillation based IDR estimation and efficient SR network structure. In addition, we provide visual comparison in Fig. 4. We can see that KDSR$_S$ has sharper edges, more realistic details, and fewer artifacts compared with other methods. More visual results are given in appendix.

Table 3: $4\times$ SR quantitative comparison on real-world SR competition benchmarks. The FLOPs and runtime are computed based on an LR size of $180 \times 320$. The best results are marked in bold.

| Methods | Parms (M) | FLOPs(G) | Runtime (ms) | AIM2019 | | | NTIRE2020 | | |
|---|---|---|---|---|---|---|---|---|---|
| | | | | LPIPS↓ | PSNR↑ | SSIM↑ | LPIPS↓ | PSNR↑ | SSIM↑ |
| ESRGAN | 16.69 | 871.25 | 236.04 | 0.5558 | 23.17 | 0.6192 | 0.5938 | 21.14 | 0.3119 |
| BSRGAN | 16.69 | 871.25 | 236.04 | 0.4048 | 24.20 | 0.6904 | 0.3691 | 26.75 | 0.7386 |
| Real-ESRGAN | 16.69 | 871.25 | 236.04 | 0.3956 | 23.89 | 0.6892 | 0.3471 | 26.40 | 0.7431 |
| MM-RealSR | 26.13 | 930.54 | 290.64 | 0.3948 | 23.45 | 0.6889 | 0.3446 | 25.19 | 0.7404 |
| KDSR$_s$-GAN (Ours) | 18.85 | 640.84 | 154.62 | **0.3758** | **24.22** | **0.7038** | **0.3198** | **27.12** | **0.7614** |

Figure 5: $4\times$ visual comparison on real-world SR competition benchmarks.

## 4.4 EVALUATION ON REAL-WORLD SR

We further validate the effectiveness of KDSR on Real-World datasets. As described in Sec. 4.1, we introduce GAN (Goodfellow et al., 2014) and perceptual Johnson et al. (2016) loss to train our network with the same high-order complex degradation process as Real-ESRGAN (Wang et al., 2021b), obtaining KDSR$_S$-GAN. We compare our methods with the state-of-the-art GAN-based SR methods, including Real-ESRGAN, BSRGAN (Zhang et al., 2021), MM-RealSR (Mou et al., 2022), ESRGAN (Wang et al., 2018b). We evaluate all methods on the dataset provided in the challenge of Real-World Super-Resolution: AIM19 Track2 (Lugmayr et al., 2019) and NTIRE2020 Track1 (Lugmayr et al., 2020). Since AIM19 and NTIRE2020 datasets provide a paired validation set, we use the LPIPS (Zhang et al., 2018b), PSNR, and SSIM for the evaluation.

The quantitative results are shown in Tab. 3. Compared with the recent real-world SR method MM-RealSR, our KDSR$_S$-GAN performs better, only consuming about 50% runtime. In addition, KDSR$_S$-GAN outperforms SOTA real-world SR method Real-ESRGAN on LPIPS, PSNR, and SSIM, only consuming its 75% FLOPs. Furthermore, we provide qualitative results in Fig. 5. We can see that our KDSR$_S$-GAN produces more visually promising results with clearer details and textures. More qualitative results are provided in appendix.

## 5 ABLATION STUDY

**Knowledge Distillation Based Blind-SR Network.** In this part, we validate the effectiveness of the components in KDSR, such as KD and IDR-DDC (Tab. 4). KDSR$_S$4 is actually the KDSR$_S$-M adopted in Tab. 1, and KDSR$_T$ is KDSR$_T$4's corresponding teacher network. **(1)** We directly input the degradation blur kernels into the KDSR$_S$4, obtaining KDSR$_S$3. Compared with KDSR$_S$3, KDSR$_S$4 has a similar performance by estimating IDR. That demonstrates that our KDSR$_S$4 can estimate quite accurate IDR to guide Blind-SR. **(2)** We cancel the KD in KDSR$_S$4 to obtain KDSR$_S$2, which means that KDSR$_S$2 cannot learn the IDR extraction from KDSR$_T$. Comparing KDSR$_S$4 and KDSR$_S$2, we can see that the KD scheme can bring 0.42dB improvement for KDSR$_S$4, which demonstrates that KD can effectively help KDSR$_S$4 to learn the IDR extraction ability from KDSR$_T$. **(3)** Based on KDSR$_S$2, we replace the IDR-DDC in IDR-DCRB with ordinary convolution to obtain KDSR$_S$1. KDSR$_S$2 is 0.17dB higher than KDSR$_S$1, which demonstrates the effectiveness of IDR-CDC. **(4)** Besides, KDSR$_S$4 is 0.2dB lower than its teacher KDSR$_T$. That means KDSR$_S$4 cannot completely learn the IDR extraction ability from KDSR$_T$.

**The Loss Functions for Knowledge Distillation.** We explore which KD function is best to guide the KDSR$_S$ learn to directly extract the same IDR as KDSR$_T$ from LR images. Although there are some works (Gao et al., 2018; He et al., 2020) have explored the KD function for SR, they take intermediate feature maps $F \in \mathbb{R}^{C \times H \times W}$ as learning objects to compress SR models. However, we take IDR $\mathbf{D}'_\mathbf{T} \in \mathbb{R}^{4C}$ as learning objects to learn the ability to extract IDR from LR images. Therefore, we cannot directly apply these experiences from previous

Table 5: Comparison between different KD loss functions.

| Loss | PNSR (dB) |
|---|---|
| $\mathcal{L}_1$ (Eq. 9) | 25.92 |
| $\mathcal{L}_2$ (Eq. 10) | 25.88 |
| $\mathcal{L}_{kl}$ (Eq. 6) | 25.96 |

Table 4: PSNR results evaluated on Urban100 with Gaussian8 ([Gu et al., 2019](#)) kernels for $4\times$ SR. The FLOPs and runtime are both measured on an LR size of $180 \times 320$.

| Method | Oracle Degradation | KD | IDR-DDC | Param (M) | FLOPs (G) | Time (ms) | PSNR (dB) |
|---|---|---|---|---|---|---|---|
| $\text{KDSR}_T$ (Ours) | ✗ | ✗ | ✓ | 5.82 | 192.77 | 39.05 | 26.16 |
| $\text{KDSR}_S 1$ | ✗ | ✗ | ✗ | 5.58 | 293.46 | 47.80 | 25.37 |
| $\text{KDSR}_S 2$ | ✗ | ✗ | ✓ | 5.80 | 191.42 | 38.74 | 25.54 |
| $\text{KDSR}_S 3$ | ✓ | ✗ | ✓ | 6.13 | 166.26 | 41.06 | 26.08 |
| $\text{KDSR}_S 4$ (Ours) | ✗ | ✓ | ✓ | 5.80 | 191.42 | 38.74 | 25.96 |

(a) IDR extracted by $\text{KDSR}_T$.    (b) IDR extracted by $\text{KDSR}_S$.    (c) IDR extracted by $\text{KDSR}_S$ without KD.    (d) IDR extracted by DASR.

Figure 6: Visualization of IDR with different isotropic Gaussian blur kernels $\sigma$ on various methods.

works to our model. Here, we define three classic KD functions: **(1)** We use the Kullback Leibler divergence to measure distribution similarity ($\mathcal{L}_{kl}$, Eq. 6). **(2)** We define $\mathcal{L}_1$ for optimization (Eq. 9). **(3)** Motivated by KD loss in SR model compression ([Gao et al., 2018](#)), we define $\mathcal{L}_2$ (Eq. 10).

$$\mathcal{L}_1 = \frac{1}{4C} \sum_{i=1}^{4C} |\mathbf{D}'_{\mathbf{S}}(i) - \mathbf{D}'_{\mathbf{T}}(i)|, \tag{9}$$

$$\mathcal{L}_2 = \frac{1}{4C} \sum_{i=1}^{4C} \left(\mathbf{D}'_{\mathbf{S}}(i) - \mathbf{D}'_{\mathbf{T}}(i)\right)^2, \tag{10}$$

where $\mathbf{D}'_{\mathbf{T}}$ and $\mathbf{D}'_{\mathbf{S}} \in \mathbb{R}^{4C}$ are IDRs extracted by $\text{KDSR}_T$-M and $\text{KDSR}_S$-M respectively. We apply these three loss functions on $\text{KDSR}_S$-M separately to learn the IDR from $\text{KDSR}_T$-M. Then, we evaluate them on $4\times$ Urban100 with Gaussian8 kernels. The results are shown in Tab. 5. We can see that the performance of $\mathcal{L}_{kl}$ is better than $\mathcal{L}_1$ and $\mathcal{L}_2$. That means that the degradation information is mainly contained in the distribution of IDR $\mathbf{D}$ rather than in its absolute values.

**The Visualization of KD-IDE.** To further validate the effectiveness of our KD-IDE, we use t-SNE ([Van der Maaten & Hinton](#), 2008) to visualize the distribution of extracted IDR. Specifically, we generate LR images from BSD100 ([Martin et al.](#), 2001) with different isotropic Gaussian kernels and feed them to $\text{KDSR}_T$, $\text{KDSR}_S$, $\text{KDSR}_S$ without KD, and DASR ([Wang et al.](#), 2021a) to generate IDR $\mathbf{D}$ for Fig. 6 (a), (b), (c), and (d) respectively. We can see from Fig. 6 (a) and (b) that $\text{KDSR}_T$ can distinguish different degradations, and $\text{KDSR}_S$ also learn this ability from $\text{KDSR}_T$ well. In addition, comparing Fig. 6 (b) and (c), we can see that $\text{KDSR}_S$ obtaining IDR extraction knowledge from $\text{KDSR}_T$ can distinguish various degradations better than $\text{KDSR}_S$ without adopting KD. That further demonstrates the effectiveness of our KD-IDE. Furthermore, we compare $\text{KDSR}_S$ and DASR (Fig. 6 (b) and (d)), and the results show that $\text{KDSR}_S$ can distinguish various degradations more clear than DASR, which shows the superiority of KD based IDE to metric learning based IDE.

## 6   CONCLUSION

Most Blind-SR methods tend to elaborately design an explicit degradation estimator for a specific type of degradation to guide SR. Nevertheless, it is difficult to provide the labels of multiple degradation combinations to train explicit degradation estimators, and these specific designs for certain degradation make them hard to transfer to other degradation processes. To address these issues, we develop a knowledge distillation based Blind-SR (KDSR) network, consisting of a KD-IDE and an efficient SR network that is stacked by IDR-DCRBs. We use KD to make KD-$\text{IDE}_S$ directly extract the same accurate IDR as KD-$\text{IDE}_T$ from LR images. IDR-DCRBs of SR network use IDR based depthwise dynamic convolution to fully and efficiently utilize the extracted IDR to guide SR. Extensive experiments on classic and complex real-world degradation processes demonstrate that the proposed KDSR can achieve a general state-of-the-art Blind SR performance.

## ACKNOWLEDGMENTS

This work was partly supported by the Alexander von Humboldt Foundation, the National Natural Science Foundation of China(No. 62171251), the Natural Science Foundation of Guangdong Province(No.2020A1515010711), the Special Foundations for the Development of Strategic Emerging Industries of Shenzhen(Nos.JCYJ20200109143010272 and CJGJZD20210408092804011) and Oversea Cooperation Foundation of Tsinghua.

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
