# OpenReview forum: "Knowledge Distillation based Degradation Estimation for Blind Super-Resolution"
_ICLR.cc/2023/Conference — ICLR 2023 poster_

### Official Review · Reviewer_GVEj · 2022-10-14

**Confidence:** 5
**Correctness:** 3
**Technical Novelty And Significance:** 2
**Empirical Novelty And Significance:** 3
**Recommendation:** 6

**Clarity, Quality, Novelty And Reproducibility:**

This paper is easy to read. The novelty of the proposed method is somehow marginal. The only new thing is implicitly estimating degradation.

**Strength And Weaknesses:**

Strength:
The idea that implicitly estimating degradation and then using it for blind SR is reasonable.

Weaknesses:
1. Estimating a degradation and then using it for blind SR is not new, the novelty of the proposed method resides in estimating it implicitly. However, most of the experiments only consider blur kernel, SR and noises. They are not sufficient enough to validate the advantages of the proposed method over existing arts. Since these degradations can be easily simulated and the settings are easily treated as GT to be learned from a network and used in the following SR network (as explicit degradation). More complicated degradations should be considered to validate the effectiveness of the proposed method.
2. According to the results of Table 3, the proposed method does not have obvious advantages relative to other methods (especially BSRGAN).
3. For Table1 and Table2, are the compared methods trained on the same training set of the proposed method? If not, the comparison is unfair.

**Summary Of The Paper:**

This paper proposes a method for blind SR in which a teacher network is used to estimate degradation from LR and HR to guide the student network to get the degradation from just LR. The experiment results show the proposed method performs better than existing methods (especially for the synthetic experiments).

**Summary Of The Review:**

The novelty is somehow marginal and the experiments cannot fully demonstrate the necessity of the proposed method.

After rebuttal, the authors address most of my concerns. I lean to accept this paper as long as the authors add more experiments and explain the reason that the proposed method has good generalization ability.

---

> ### Author Response · Authors · 2022-11-16
> **Author feedback to Reviewer GVEj**
>
> **Q1: More complicated degradations should be considered to validate the effectiveness of the proposed method.**
> A1: Thanks for your suggestions! In Table 3, we explore our KDSR on Real-world SR as Real-ESRGAN, in which LR images contain very complex degradations, including blur, noise, down-sampling, JPEG compression, etc. Furthermore, to better simulate the complex degradations and cover degradation space in the real world, our KDSR and Real-ESRGAN also apply a random shuffle of degradation orders and second-order degradation.
> If the reviewer has any further suggestions on degradation choice, we are happy to do additional experiments to test it.
> ___
> **Q2:According to the results of Table 3, the proposed method does not have obvious advantages relative to other methods (especially BSRGAN).**
>
> A2: Our improvement is significant. GAN based SR methods put more emphasis on LPIPS which can better reflect perceptual quality compared with PSNR. For LPIPS,  MM-RealSR (ECCV2022) surpasses Real-ESRGAN (ICCVW21) by 0.0008 and 0.0025 on AIM2019 and NTIRE2020, respectively. Our KDSRs-GAN can surpass MM-RealSR by  0.0209 and 0.0247 on AIM2019 and NTIRE2020, respectively, while only consuming almost half the inference time of MM-RealSR. Besides, the visual quality of our KDSRs-GAN (Fig.5, Fig 10) is also better than other Real-World SR methods.
>
> ___
>
> **Q3:For Table1 and Table2, are the compared methods trained on the same training set of the proposed method? If not, the comparison is unfair.**
>
> A3: Yes, we train our method with the same settings as comparing methods.

---

> > ### Comment · Reviewer_GVEj · 2022-11-18
> > **remaining issues**
> >
> > For additional datasets:
> >
> > The provided 'real-world SR' datasets (AIM19 Track2 and NTIRE2020 Track1) are not real 'real-world' SR datasets which is also pointed out by other reviewers (cRmZ and HYPQ). More datasets should be tested quantitatively and qualitatively such as [1], [2] and [3]. If the authors only test their method on synthetic data and they train their network in a dataset generated in a similar way, it is difficult to demonstrate the advantage that estimating an implicit degradation for real SR.
> >
> > [1] Designing a Practical Degradation Model for Deep Blind Image Super-Resolution
> >
> > [2] Component Divide-and-Conquer for Real-World Image Super-Resolution
> >
> > [3] DSLR-Quality Photos on Mobile Devices with Deep Convolutional Networks
> >
> > For the advantage relative to other methods:
> >
> > The improvement is not 'significant'. I admit that LPIPS is a good metric to measure perceptual quality. However, perceptual quality and PSNR are like a trade-off in SR (which is demonstrated in Fig 2 of [4]). Compared with BSRGAN, the proposed method has better LPIPS and worse PSNR. In this way, it is difficult to tell which method is better. These are also pointed out by y283.
> >
> > [4] ESRGAN: Enhanced Super-Resolution Generative Adversarial Networks
> >
> > Another issue is how the proposed knowledge distillation can estimate the degradation without introducing other information such as high-resolution textures. For the teacher network, its input contains high-resolution images and the generated features can help the student network to exact high-resolution textures. Even though Figure 6 experimentally demonstrates the knowledge distillation can estimate the degradation, more detailed analysis is still needed.

---

> > > ### Author Response · Authors · 2022-11-19
> > > **Author feedback to Reviewer GVEj**
> > >
> > > **Q4: The provided 'real-world SR' datasets (AIM19 Track2 and NTIRE2020 Track1) are not real 'real-world' SR datasets which is also pointed out by other reviewers (cRmZ and HYPQ). More datasets should be tested quantitatively and qualitatively such as [1], [2] and [3]. If the authors only test their method on synthetic data and they train their network in a dataset generated in a similar way, it is difficult to demonstrate the advantage that estimating an implicit degradation for real SR.**
> > >
> > > A4: Thanks for your suggestion!
> > >
> > > (1) We would like to clarify that we have already included evaluation results on “real-world” SR datasets (NTIRE2020 track2), which were captured using smartphones (please kindly see Figs 7 and 8 in the revised paper and whole images in supplementary material). The experimental comparison shows that our method outperforms recent SOTA blind SR approaches. Reviewer HYPQ thinks our persuasive experimental results can address his concern.
> > >
> > > (2) We further conduct experiments on other “real-world” SR datasets (DRealSR datasets).  please kindly see Fig 9 in the revised paper. The experimental comparison shows that our KDSRs-GAN achieves better visual quality than compared SOTA blind SR methods.
> > >
> > > (3) Besides, the degradation in test datasets: AIM2019 Track2 and NTIRE2020 Track1 are unknown.  We train our KDSRs-GAN with the same degradation settings as Real-ESRGAN, which is different from test datasets. Thus, the superior results of our model can demonstrate the generalization ability of the proposed KDSRs-GAN.
> > >
> > > ___
> > >
> > > **Q5: Compared with BSRGAN, the proposed method has better LPIPS and worse PSNR. In this way, it is difficult to tell which method is better. These are also pointed out by y283.**
> > >
> > > A5: Thanks for your suggestion! First, we want to certify that we did not make any trade-off to increase our LPIPS results. The BSRGAN and Real-ESRGAN focus on designing image degradation models. Note that they have the same blind SR network architecture.  BSRGAN achieves higher PSNR than our method and Real-ESRGAN because it learned from a different degradation with Real-ESRGAN.  In our work, we aim to develop a new blind SR approach and directly use the degradation model from Real-ESRGAN, which can generate more diverse and realistic degradations for better visual quality. Compared to the Real-ESRGAN, our method achieves both better PSNR and LPIPS consuming fewer FLOPs, which can demonstrate the effectiveness of our blind SR approach.
> > >
> > > ___
> > >
> > > **Q6:Another issue is how the proposed knowledge distillation can estimate the degradation without introducing other information such as high-resolution textures. Even though Figure 6 experimentally demonstrates the knowledge distillation can estimate the degradation, more detailed analysis is still needed.**
> > >
> > > A6: Thanks for your suggestion!  Reviewer HYPQ raised a similar concern in the first round. We are happy to see that our answer has addressed his question. But, to further clarify the concern, we would like to provide more additional experiments and analyses.
> > >
> > >
> > > We further conduct experiments on Set5  to demonstrate that our KD-IDE can extract the degradation. Specifically, we first generated a LR image (Index = 1) using a Gaussian kernel $\sigma=2.6$. Then, we randomly selected another 4 HR images to generate LR images (Index = 2, 3, 4, 5) with the same Gaussian kernel $\sigma=2.6$. Besides, for index=6, we use the same HR image as index=1 to generate LR images with Gaussian kernel $\sigma=3.2$. Then, we extract degradation representations from the 6 images to super-resolve the image with 1 index individually. The results are shown in the following table. Comparing idx=1,2,3,4,5, we can see that our method is relatively stable to content variation and textures. Comparing idx=1,6, we can see that the extracted degradation by our KDSR is far more relevant to degradation rather than content and textures.
> > >
> > > Please let us know if the reviewer wants to see any other experiments and analyses.
> > >
> > > | Index | 1     | 2     | 3     | 4     | 5     | 6      |   |   |   |
> > > |-------|-------|-------|-------|-------|-------|--------|---|---|---|
> > > | PSNR  | 27.55 | 27.54 | 27.53 | 27.54 | 27.54 | 26.44  |   |   |   |
> > > |       |       |       |       |       |       |        |   |   |   |

---

> > > > ### Comment · Reviewer_GVEj · 2022-11-30
> > > > **remaining issues**
> > > >
> > > > For A4:
> > > >
> > > > For real-world SR datasets, please provide quantitative results other than some selected images. And more datasets still need to be tested.
> > > >
> > > > For generalization ability, please demonstrate why the proposed method can work well under unseen degradations.
> > > >
> > > > For A5:
> > > >
> > > > The trade-off means that the proposed method will have a similar LPIPS compared with BSRGAN if the proposed method and BSRGAN have similar PSNR. In this way, it seems the proposed method does not have significant improvement. But I think the proposed method does provide a new idea to deal with unknown degradations.
> > > >
> > > > In general, I think this paper has some merits. My major concern is the authors should test their method under more real-world sr datasets and demonstrate why the proposed method can generalize well on unseen degradations.

---

> > > > > ### Author Response · Authors · 2022-12-01
> > > > > **Author feedback to Reviewer GVEj**
> > > > >
> > > > > Thanks for the new response! We appreciate that the reviewer acknowledges the novelty of our idea!
> > > > >
> > > > > ----
> > > > >
> > > > > **Q7:The trade-off means that the proposed method will have a similar LPIPS compared with BSRGAN if the proposed method and BSRGAN have similar PSNR. In this way, it seems the proposed method does not have significant improvement. But I think the proposed method does provide a new idea to deal with unknown degradations.**
> > > > >
> > > > >
> > > > > A7: Thanks for your suggestions!
> > > > >
> > > > > To further support our previous arguments (A4) about the trade-off issue raised by the reviewer, we further use BSRGAN’s degradation model to train our KDSR-GAN. As shown in the following table, our model (last row)  can achieve better PSNR and  LPIPS consuming even fewer FLOP than BSRGAN. The results can validate that there is no trade-off between LPIPS and PSNR in our approach.
> > > > >
> > > > > |                                             |          |        | AIM2019 |        |        | NTIRE2020 |         |   |   |
> > > > > |---------------------------------------------|----------|--------|---------|--------|--------|-----------|---------|---|---|
> > > > > |                                             | FLOPs(G) | LPIPS  | PSNR    | SSIM   | LPIPS  | PSNR      | SSIM    |   |   |
> > > > > | Real-ESRGAN                                 | 236.04   | 0.3956 | 23.89   | 0.6892 | 0.3471 | 26.40      | 0.7431  |   |   |
> > > > > | Ours (using Real-ESRGAN degradation model ) | 154.62   | 0.3739 | 24.10    | 0.6946 | 0.3199 | 26.66     | 0.7487  |   |   |
> > > > > | BSRGAN                                      | 236.04   | 0.4048 | 24.20    | 0.6904 | 0.3691 | 26.75     | 0.7386  |   |   |
> > > > > | Ours (using BSRGAN degradation model )      | 154.62   | 0.3921 | 24.26   | 0.6970  | 0.3382 | 27.24     | 0.7529  |   |   |
> > > > > |                                             |          |        |         |        |        |
> > > > >
> > > > > ----
> > > > >
> > > > > **Q8: For real-world SR datasets, please provide quantitative results other than some selected images. And more datasets still need to be tested.**
> > > > >
> > > > > A8: Thanks for your suggestion! We further test our method on RealSR and DRealSR.  We can see that our methods significantly surpass corresponding real-world SR methods with the same degradation model on LPIPS, PSNR, and SSIM consuming fewer FLOPs.
> > > > >
> > > > > |                                                         |        | RealSR |        |        | DRealSR |         |   |   |   |
> > > > > |---------------------------------------------------------|--------|--------|--------|--------|---------|---------|---|---|---|
> > > > > |                                                         | LPIPS  | PSNR   | SSIM   | LPIPS  | PSNR    | SSIM    |   |   |   |
> > > > > | BSRGAN                                         | 0.3648 | 26.90   | 0.7912 | 0.3937 | 28.21   | 0.8085  |   |   |   |
> > > > > | Ours (using BSRGAN degradation model )                  | 0.3584 | 27.36  | 0.8049 | 0.3702 | 28.45   | 0.8261  |   |   |   |
> > > > > | Real-ESRGAN                                  | 0.3629 | 26.07  | 0.7864 | 0.3800   | 27.98   | 0.8127  |   |   |   |
> > > > > | MM-RealSR (using Real-ESRGAN degradation model) | 0.3606 | 24.07  | 0.7708 | 0.3773 | 26.27   | 0.8028  |   |   |   |
> > > > > | Ours (using Real-ESRGAN degradation model )             | 0.3579 | 26.72  | 0.7987 | 0.3688 | 28.16   | 0.8169  |   |   |   |
> > > > > |                                                         |        |        |        |        |
> > > > >
> > > > >
> > > > > ----
> > > > >
> > > > > **Q9:For generalization ability, why the proposed method can work well under unseen degradations.**
> > > > >
> > > > >
> > > > >
> > > > > A9: Real-ESRGAN constructs a complex enough degradation model for covering the real-world degradation space, which enables model generalization. But, due to the complexity of the degradation model, it is hard to explicitly learn a degradation estimator with specific labels. To solve the issue, our approach learns an implicit degradation estimator without relying on specific degradation labels. It can extract latent degradation representation to guide SR, which can help to better capture and adapt to complex degradations constructed by the Real-ESRGAN’s degradation model. Thus, it is more capable of handling real-world and even unseen degradations in natural images.

---

> ### Author Response · Authors · 2022-11-30
> **Further discussion with Reviewer GVEj**
>
> Dear Reviewer GVEj,
>
> We thank you for the precious review time and valuable comments. We have referred to your remaining suggestions and made the following improvements in the revised paper:
>
> **(1)** We included extensive evaluation results on “real-world” SR datasets (NTIRE2020 track2 and DRealSR), which were captured using smartphones (please kindly see Figs 7, 8 and 9 in the revised paper and whole images in supplementary material). The visual comparison shows that our KDSR-GAN can significantly surpass other SOTA real-world SR methods, such as MM-RealSR (ECCV2022), which is also acknowledged by reviewer HYPQ.
>
> **(2)** We clarify the effectiveness of our KDSR. Our KDSR adopts the degradation model from Real-ESRGAN (ICCVW21). Compared to the Real-ESRGAN and MM-RealSR (ECCV2022, also adopt the degradation model from Real-ESRGAN), our method achieves better  LPIPS, PSNR, and SSIM consuming fewer FLOPs. Furthermore, extensive visual comparisons in A4 also demonstrate that our KDSR can have much better visual quality than other Real-World SR methods (BSRGAN, Real-ESRGAN, MM-RealSR).
>
> **(3)** We provide more additional experiments and analyses to demonstrate the effectiveness of our KD-IDE, which is also acknowledged by reviewer HYPQ.
>
> We believe our responses have covered your remaining concerns and hope to discuss further with you whether or not your concerns have been addressed.  Please let us know if you still have any unclear parts of our work.
>
> Best,
>
> Authors

---

> ### Author Response · Authors · 2022-12-03
> **Further discussion with Reviewer GVEj (2)**
>
> Dear Reviewer GVEj,
>
> We thank you for the precious review time and valuable comments. We also appreciate that the reviewer acknowledges the novelty and merits of our paper! We have referred to your recent remaining suggestions and made the following improvements:
>
> (1) We further use BSRGAN’s degradation model to train our KDSR-GAN to support our arguments (A4) about the trade-off issue. The results show that our model can achieve better PSNR and LPIPS consuming even fewer FLOP than BSRGAN. The results can validate that there is no trade-off between LPIPS and PSNR in our approach and our improvement is significant (A7).
>
> (2) We further test our method on more real-world datasets, such as RealSR and DRealSR. We can see that our methods significantly surpass corresponding real-world SR methods with the same degradation model on LPIPS, PSNR, and SSIM consuming fewer FLOPs ( A8 ).
>
> (3) We clarify the reason of our better  generalization ability of handling real-world and even unseen degradations in natural images.
>
> We believe our responses have covered your recent remaining concerns and hope to discuss further with you whether or not your concerns have been addressed. Please let us know if you still have any unclear parts of our work.
>
> Best,
>
> Authors

---

> ### Author Response · Authors · 2022-12-09
> **Thanks Reviewer GVEj for approving our work**
>
> Dear Reviewer GVEj,
>
> Thanks for your approving our work. We are happy to see that our experimental results and response can solve your concerns. We also appreciate that you acknowledge our reasonable implicit degradation estimator design, novelty, and good generalization ability.
>
> Best,
>
> Authors

---

### Official Review · Reviewer_HYPQ · 2022-10-23

**Confidence:** 5
**Correctness:** 4
**Technical Novelty And Significance:** 3
**Empirical Novelty And Significance:** Not applicable
**Recommendation:** 6

**Clarity, Quality, Novelty And Reproducibility:**

The idea is interesting and clearly illustrated. The provided details seem to be adequate enough to re-implement the experiments.

**Strength And Weaknesses:**

1.  The proposed method is novel and performs well on synthetic datasets.

2. The authors claim that the proposed KDSR implicitly estimates a degradation representation. While there are no experiments analyzing the estimated representations. The authors are suggested to compare the estimated degradations of images with the same/similar/different degradations. If the authors' claim stands, images that have different content but the same degradations will also have a similar estimated representation.

3. The experiments are performed on synthetic datasets. The authors are suggested to verify their methods on real-image datasets, such as the track2 of NTIRE2020.

4. It will be better if the visual results provide comparisons of the whole images, since the degradation of the LR image may be spatially variant, while the estimated degradation representation is spatially invariant.

**Summary Of The Paper:**

This paper proposes to use a knowledge distillation based to learn an implicit representation of degradations. The proposed method can handle complex degradation and does not need explicit degradation supervision. The proposed method finally achieve better results on several synthesized datasets.

**Summary Of The Review:**

This paper proposes a novel idea to implicitly estimate the degradation. While the experiments do not fully explore the estimated degradations and verify the method on real-image datasets. I will consider changing my opinion if the authors can provide more experiment details in the rebuttal phase.

--Post Rebuttal--:
The authors provide more persuasive experimental results and the response addresses most of my concerns. I would like to improve my score.

---

> ### Author Response · Authors · 2022-11-16
> **Author feedback to Reviewer HYPQ**
>
> **Q1:The authors are suggested to compare the estimated degradations of images with the same/similar/different degradations. If the authors' claim stands, images that have different content but the same degradations will also have a similar estimated representation.**
>
> A1:Thanks for your suggestion! We think Fig.6 (b) of our paper can address your concern.  Specifically, we apply 5 isotropic Gaussian kernels on BSD100 separately to obtain LR images. Then, we feed these LR images to
> KDSRs to obtain corresponding implicit degradation representation (IDR). After that, we use T-SNE to visualize the IDR. We can see that, in Fig.6 (b), the IDR extracted from the same degradation on different images will have similar distribution and be clustered together.
>
> ___
>
> **Q2:The experiments are performed on synthetic datasets. The authors are suggested to verify their methods on real-image datasets, such as the track2 of NTIRE2020.**
>
> A2：Thanks for your suggestion! We have added the comparison on track2 of NTIRE2020 in Figs. 7 and 8 of the revised paper. We can see that our KDSRs-GAN has better visual quality than other compared methods.
>
> ___
>
> **Q3:It will be better if the visual results provide comparisons of the whole images, since the degradation of the LR image may be spatially variant, while the estimated degradation representation is spatially invariant.**
>
>
> A3: (1) Thanks for your suggestion! Since the restored images are quite large, we included comparison results on whole images in the supplementary material.  We can see that our Implicit degradation estimation of spatially invariant can produce quite good visual results.

---

> ### Author Response · Authors · 2022-11-30
> **Thanks Reviewer HYPQ for approving our work**
>
> Dear Reviewer HYPQ,
>
> Thanks for your approving our work. We are happy to see that our experimental results and response can solve your concerns. We also appreciate that you acknowledge our novelty, persuasive experimental results,  presentation, good performance, and reproducibility
>
> Best,

---

### Official Review · Reviewer_cRmZ · 2022-10-24

**Confidence:** 4
**Correctness:** 4
**Technical Novelty And Significance:** 3
**Empirical Novelty And Significance:** 3
**Recommendation:** 6

**Clarity, Quality, Novelty And Reproducibility:**

The idea is neat and the paper presents a method that produces nice results (on the shown benchmarks). The presentation is a little disorganized and hard to follow, which hinders the contributions and make it hard to analyze the real novelty and contributions.

**Strength And Weaknesses:**

Strengths:
1. The paper introduces an interesting idea to address a highly relevant practical problem (blind real image super-resolution)
2. The paper presents many experiments and ablation studies showing that the method is robust and produces state-of-the-art results in terms of distortion metrics (PSNR).
3. The method is well detailed, and well motivated.

Weakness:
1. Presentation is cumbersome. It is hard to disentangle the contributions of the paper since everything is mixed together. Details about the architecture are given as long as the mathematical formulation. It would much better if the paper first present a mathematical formulation of the problem (the training, the loss functions, the goals), and later present the architecture and the details about the modules. In the current formulation is extremely hard to distill the ideas and understand what is new and what is not.
2. All the results are with benchmarks that use simulated degradations. It would be nice to see results (visual) on real images captured by any camera since the paper is claiming that addresses the real image super-resolution problem.


**Summary Of The Paper:**

This paper introduces a method for blind single image super-resolution. Contrary to the mainstream the idea of this method is to design an implicit degradation estimator that can extract information regarding the degradation without having an explicit model. This is done using ideas from Knowledge distillation where first a teacher model is trained using paired LR/HR images and then a student model is trained only using LR images to predict the same degradation representation.

**Summary Of The Review:**

This paper presents an interesting idea of using knowledge distillation to learn the degradation operation on the blind single image super-resolution task. The method produces very good results in the shown benchmarks. The main issue is the presentation. It is hard to disentangle the mathematical formulation from the implementation as everything is mixed together. Nonetheless I think this paper introduced a novel idea so I'm leaning towards accepting the paper. But, I would like to see a better presentation where the mathematical formulation (e.g., the abstract description of the modules, the goals of each module, the loss functions) is given in a separate section as the details about the architecture (and the exact implementation of the modules). I would like to see also results on real images (not from a benchmark). For example results on any image of the Open Image Dataset and a visual comparison in terms of artifacts (since no metric will be available).

After Rebuttal.
The authors addressed the concerns raised during the first round of reviews, and updated the manuscript accordingly. I'm leaning towards acceptance and thus I keep my original score.

---

> ### Author Response · Authors · 2022-11-16
> **Author feedback to Reviewer cRmZ**
>
> **Q1: Presentation is cumbersome.  It would much better if the paper first present a mathematical formulation of the problem (the training, the loss functions, the goals), and later present the architecture and the details about the modules.**
>
> A1: Thanks for your suggestion! We have added an overview at the beginning of Sec.3 in the revised paper to introduce the mathematical formulation of the problem.
>
> ___
>
> **Q2:All the results are with benchmarks that use simulated degradations. It would be nice to see results (visual) on real images captured by any camera since the paper is claiming that addresses the real image super-resolution problem.**
>
>
> A2: Thanks for your suggestion! We have added results of real images from cameras in Figs. 7 and 8 of the revised paper. We can see that our KDSRs-GAN has better visual quality than other compared methods. Moreover, we included visual comparison results on whole images from cameras in the supplementary material.

---

> ### Author Response · Authors · 2022-12-03
> **Thanks Reviewer cRmZ for approving our work**
>
> Dear Reviewer cRmZ,
>
> Thanks for your approving our work. We are happy to see that our experimental results and responses can solve your concerns. We also appreciate that you acknowledge our novelty, experiments, very good results,  and motivation.
>
> Best,
>
> Authors

---

### Official Review · Reviewer_y283 · 2022-11-03

**Confidence:** 4
**Correctness:** 3
**Technical Novelty And Significance:** 2
**Empirical Novelty And Significance:** 2
**Recommendation:** 6

**Clarity, Quality, Novelty And Reproducibility:**

The paper is nicely structured and the presentation is clear. The experimental details are provided along with the source code to foster reproducibility. See main review for quality and novelty of the work.


**Strength And Weaknesses:**

**Main Review**

**Major Strengths**

Real-world blind super-resolution with a combination of multiple unknown degradation is a challenging problem to deal with using explicit degradation models. This paper proposes to learn an implicit degradation model by leveraging a previously known method called knowledge distillation in a teacher-student setup.

***Experiments***

The experiments are comprehensive and provide comparable or better performance than compared methods. The proposed method is computationally efficient in general.

Experiments on various datasets with different degradation models show promising results.

The authors demonstrate generalization to several degradations which is an important highlight of this work.

**Weaknesses**

The main idea is an amalgamation of previously known methods, such as knowledge distillation and efficient super-resolution using GANs.

The loss function consists of standard objective functions with several regularization parameters. The authors should address how to tune all these parameters in large-scale real-world blind super-resolution tasks.

Table 3: Are the compared methods really the state-of-the-art for real-world SR on AIM19 and NTIRE2020?

In the literature, the results of DASR on AIM19 are reported to be better than the results obtained by the proposed method KDSR. In my opinion, the authors should clarify how the compared methods are SOTA, maybe in terms of the training procedure, e.g., paired or unpaired learning or appropriate metrics to make a proper assessment of the baselines.

There is a disparity between the perceptual metrics, such as FID and LPIPS of popular SR methods including DASR (also the winner of AIM19 challenge FSSR) and the distortion metrics, such as PSNR and SSIM. As a result, there is a trade-off between perceptual and distortion quality. I wonder whether the disparity exists in the proposed method KDSR. If yes, how to overcome such issues?

It is well-known that quantitative improvements in terms of PSNR and SSIM do not necessarily indicate qualitative improvements, especially for the tasks under consideration. This is evident from Table 1, 2, 3, 4, and 5 of the main paper because most methods have similar scores. To make a fair comparison with baselines, I suggest the authors include LPIPS, FID or other image quality metrics on all the datasets.

Figure 8: Are the results generated for isotropic or non-isotropic Gaussian kernels?

**Summary Of The Paper:**

Most of the prior works for blind image super-resolution design explicit degradation estimation, which is often infeasible due to lack of proper degradation labels. Also, there might not exist a universal degradation model to capture various practical degradation types, such as blur, noise, compression, and platform instability which makes this problem particularly interesting. This paper proposes a method to learn an implicit degradation model leveraging learned representation in a teacher-student model. One of the important highlights of this paper is the generalization to various (real-world) degradation settings. Experiments show comparable or better performance than some existing models on commonly used datasets. The experiments on AIM19 and NTIRE2020 do not offer a significant gain over prior works which have not been included due to some reasons. The proposed approach is an amalgamation of previously known methods in related disciplines.


**Summary Of The Review:**

This paper proposes a method to learn an implicit degradation model leveraging learned representation in a teacher-student model. One of the important highlights of this paper is the generalization to various (real-world) degradation settings. Experiments show comparable or better performance than some existing models on commonly used datasets. The experiments on AIM19 and NTIRE2020 do not offer a significant gain over prior works which have not been included due to some reasons. The proposed approach is an amalgamation of previously known methods in related disciplines.

===========================================
I thank the authors for their detailed response. I think the paper is in a good shape now, and it'd be interesting to many at the conference. Based upon the discussion, I plan to rate this as marginally above the acceptance level.

---

> ### Author Response · Authors · 2022-11-16
> **Author feedback to Reviewer y283 (Part1)**
>
> **Q1: The main idea is an amalgamation of previously known methods, such as knowledge distillation and efficient super-resolution using GANs.**
>
> A1: (1) Knowledge distillation is a general concept for model compression. Rather than solving model compression, we adopt the KD to design a novel implicit degradation estimator (KD-IDE) for blind SR.
>
>  (2) We would like to note that the idea of learning implicit degradation is new for blind SR. KD is adopted as an approach to implement the idea. Explicit degradation estimations are complex, time-consuming, and hard to obtain precise degradation ground-truth labels for training.  Moreover, previous Blind-SR models mostly need to design a specific explicit degradation estimator for different degradation settings. Our implicit degradation estimator does not need degradation labels for training and can generalize to any degradation process and achieve better performance, which is simple, strong, and general and has the potential to be the backbone to realize the unification of blind-SR.
>
> (3) Our efficient super-resolution framework is also meaningful and novel. It can fully and efficiently utilize the implicit degradation representation to guide SR. Moreover, its also neat structure is very suitable for deployment.
>
> ___
>
> **Q2:The loss function consists of standard objective functions with several regularization parameters. The authors should address how to tune all these parameters in large-scale real-world blind super-resolution tasks.**
>
> Thanks for your suggestions! For the real-world SR, we try various regularization parameter settings on $\lambda_{rec}$, $\lambda_{per}$, and $\lambda_{adv}$ and find that our KDSR-GAN can have the best visual results as $\lambda_{rec}=1$, $\lambda_{per}=1$ and $\lambda_{adv}=0.1$. It is notable that previous Real-SR works also adopt the same hyperparameter settings to balance the perception and distortion. For the $\lambda_{kl}$, we also try several settings and find $\lambda_{kl}=1$ can balance the implicit degradation learning and image restoration learning. Overall, these hyperparameter settings are robust.
> ___
>
> **Q3: Table 3: Are the compared methods really the state-of-the-art for real-world SR on AIM19 and NTIRE2020?**
>
> A3: Yes, they are recent state-of-the-art real-world SR approaches, such as BSRGAN (ICCV21), Real-ESRGAN (ICCVW2021), and MM-RealSR [1] (ECCV2022). Please see more details in A4.
>
>
> [1] Chong Mou, Yanze Wu, Xintao Wang, Chao Dong, Jian Zhang, and Ying Shan. Metric learning based interactive modulation for real-world super-resolution.ECCV, 2022.
> ___
>
> **Q4:In the literature, the results of DASR on AIM19 are reported to be better than the results obtained by the proposed method KDSR. In my opinion, the authors should clarify how the compared methods are SOTA, maybe in terms of the training procedure, e.g., paired or unpaired learning or appropriate metrics to make a proper assessment of the baselines.**
>
> A4: (1) Thanks for your suggestion! DASR is excellent work. However, DASR merely extracts limited degradations in AIM19  to train networks for better competition performance, which cannot cover far more complex and diverse real-world degradations. Our KDSR-GAN adopts the complex degradation training settings of Real-ESRGAN. Specifically, the training degradation of our KDSR and Real-ESRGAN is far more complex than DASR. We introduce comprehensive degradation operations such as blur, noise, down-sampling, and JPEG compression, and control the severity of each operation by randomly sampling the respective hyper-parameters. Furthermore, to better simulate the complex degradations and cover degradation space in the real world, our KDSR and Real-ESRGAN also apply a random shuffle of degradation orders and second-order degradation.
>
> (2) Following previous works, such as MM-RealSR (ECCV2022), we just compare methods trained with complex degradations.
>
>
> ___
>
> **Q5:There is a disparity between the perceptual metrics and the distortion metrics. I wonder whether the disparity exists in the proposed method KDSR. If yes, how to overcome such issues?**
>
> A5: The disparity exists in almost all SR methods, including our KDSR. For the classic degradation problems (blur and noise), as previous works did, we use L1 loss to make networks obtain higher PSNR/SSIM, which can represent the learning ability of networks. For real-world SR, we pursue more perceptual quality and introduce adversarial loss and perceptual loss. However, perceptual quality is hard to be measured by metrics. Therefore, we adjust the hyperparameters of  $\lambda_{rec}$, $\lambda_{per}$, and $\lambda_{adv}$  and empirically select one with the best visual quality.

---

> ### Author Response · Authors · 2022-11-16
> **Author feedback to Reviewer y283 (Part2)**
>
> **Q6:To make a fair comparison with baselines, I suggest the authors include LPIPS, FID or other image quality metrics on all the datasets.**
>
> A6: Thanks for your suggestions! We add LPIPS and FID comparisons in Tables 6,7,8,9, and10 of the revised paper.
> We can see that our approach outperforms all of the compared methods in terms of LPIPS and FID.
> ___
>
> **Q7:Figure 8: Are the results generated for isotropic or non-isotropic Gaussian kernels?**
>
> A7:The results are generated for the non-isotropic Gaussian kernels and noises.

---

> ### Author Response · Authors · 2022-12-03
> **Thanks Reviewer y283 for approving our work**
>
> Dear Reviewer y283,
>
> Thanks for your approving our work. We are happy to see that our experimental results and responses can solve your concerns. We also appreciate that you acknowledge our experiments, promising results, generalization, novelty, clear presentation, reproducibility and paper shape.
>
> Best,
>
> Authors

---

### Author Response · Authors · 2022-11-21
**Response to all reviewers and area chairs for a brief summary**

Dear Reviewers,

We would like to thank all reviewers for their detailed and valuable comments. We have responded to each reviewer individually to address any comments. Meanwhile, the manuscript has been updated. We would like to give a brief summary here.

+ We further added LPIPS and FID comparisons in Tables 6,7,8,9, and10.

+ We added the comparison on track2 of NTIRE2020 (real-world datasets, which is captured using smartphone cameras) in Figs. 7 and 8. Reviewer HYPQ thinks our persuasive experimental results can address his concern.

+ To better validate the visual quality on whole real-wolrd image captured using cameras, we provide more visual comparisons in the supplementary Material.

+  We further added the comparison on DRealSR (real-world datasets, which is captured using smartphone cameras) in Fig. 9.

+ We take valuable suggestions proposed by reviewers to make the paper more clear. Specifically,  we revised our presentation, and added an overview at the beginning of Sec.3 in the revised paper to introduce the mathematical formulation of the problem and clarify the difference between our KDSR and previous works. Note that we revise the text contents and mark them in blue. You can easily find them.

+ Except Fig.6 (b), we further conduct new experiments in A6 to Reviewer GEVj to further validate KD-IDE can estimate the degradation. Notably, Reviewer HYPQ raised a similar concern in the first round. We are happy to see that our answer has addressed his question.

The valuable suggestions of reviewers have greatly benefited our paper. We are happy to see some further discussions to address any of your concerns.

Thanks again.

---

> ### Author Response · Authors · 2022-12-03
> **Response to all reviewers and area chairs for a brief summary**
>
> Dear Reviewers,
>
> We would like to thank all reviewers for their detailed and valuable comments. Recently, we have added new experiments to better address comments from reviewer GVEj. We also appreciate that the reviewer GVEj acknowledges the novelty and merits of our paper!  We would like to give a brief summary here.
>
> + To further support our arguments (A4 to reviewer GVEj) about the trade-off issue, we further use BSRGAN’s degradation model to train our KDSR-GAN. The results show that our model can achieve better PSNR and LPIPS consuming even fewer FLOP than BSRGAN. The results can validate that there is no trade-off between LPIPS and PSNR in our approach and improvement is significant (A7 to reviewer GVEj).
>
>  + We further test our method on more real-world datasets, such as RealSR and DRealSR. We can see that our methods significantly surpass corresponding real-world SR methods with the same degradation model on LPIPS, PSNR, and SSIM consuming fewer FLOPs ( A8 to reviewer GVEj).
>
> The valuable suggestions of reviewers have greatly benefited our paper. We believe our responses have covered the recent remaining concerns and are happy to see some further discussions to address any of your concerns.
>
> Thanks again.

---

### Author Response · Authors · 2022-12-09
**Response to all reviewers and area chairs**

Dear Reviewers and area chairs,

We thank all reviewers and area chairs for their valuable time and comments. After discussing with all reviewers and providing more clarifications/results/analyses, we would like to give a brief response.

All reviewers now agree with the novelty, good results, experiments, and writing/organization of our paper.

All reviewers now hold a **positive** side for our work. Our responses have covered their questions.

We have submitted the demo code of this paper. We would make all the code, trained models, and results available to the public soon. We think that the proposed KD based implicit degradation estimation is simple, strong, and general and has the potential to be the backbone to realize the unification of blind SR and even other restoration tasks (denoising, deblocking, deblur). Besides, our SR framework can efficiently utilize the implicit degradation representation to guide restoration. Its neat and efficient structure is practical and very suitable for deployment. We hope to further investigate this direction in low-level vision together with other researchers.

We thank all reviewers and area chairs again!

Best,

Authors

---

### Decision · Program_Chairs · 2023-01-20

**Decision:**

Accept: poster

**Justification For Why Not Higher Score:**

The paper is a borderline paper, but the authors addressed all the concerns and made substantial revision on the paper by including extra experiments and following the suggestion provided by the reviewers. It is good to accept it as a poster.

**Justification For Why Not Lower Score:**

The authors addressed all the concerns and made substantial revision on the paper by including extra experiments and following the suggestion provided by the reviewers. It is good to accept it as a poster.

**Metareview: Summary, Strengths And Weaknesses:**

This paper introduces a new method for blind single image super-resolution. The authors design an implicit degradation estimator that can extract information about degradation without having an explicit model. This is done by using knowledge distillation where first a teacher model is trained using paired LR/HR images and then a student model is trained only using LR images to predict the same degradation representation.  Experiments show that the proposed method can achieve better results on several synthesized datasets. The paper received a total of 4 valid reviews. The rebuttal successfully addressed all the concerns from the reviewers, and in the final comments, all reviewers agree with the novelty, good experiments, as well as the presentation of the paper, and recommend accepting the paper. Additionally, the authors have made substantial revision on the paper by taking into account all the suggestions from the reviewers and including extra experimental results. The AC appreciates the efforts of the authors in improving the paper during the rebuttal and agrees with the judgements from the reviewers, thus recommending accepting the paper because of its novelty and solid experiments.


**Note From Pc:**

if the above contains the word "oral" or "spotlight" please see: "oral" presentation means -> notable-top-5% and "spotlight" means -> notable-top-25%. As stated in our emails, we are disassociating presentation type from AC recommendations

**Summary Of Ac-Reviewer Meeting:**

Even though the paper is a borderline paper (because its average score is 6), all 4 reviewers are satisfied with the rebuttal and agree to accept the paper. All comments and decisions made by the reviewers are very clear in the forum, so the AC thinks that there is no need for organizing an AC-reviewer meeting for final decision making.  Given the fact that the authors have addressed all the concerns and the revised paper has been improved with the help of all 4 reviewers, the AC recommends accept it as a poster.